# Factors Influencing Workplace Health Promotion Interventions for Workers in the Semiconductor Industry According to Risk Levels of Chronic Disease

**DOI:** 10.3390/ijerph182111383

**Published:** 2021-10-29

**Authors:** Yun-Kyoung Song, Boyoon Choi, Kyungim Kim, Hyun Jin Park, Jung Mi Oh

**Affiliations:** 1Research Institute of Pharmaceutical Sciences, College of Pharmacy, Seoul National University, Seoul 08826, Korea; yksong@cu.ac.kr (Y.-K.S.); boyoon1@snu.ac.kr (B.C.); hjpark059@snu.ac.kr (H.J.P.); 2College of Pharmacy, Daegu Catholic University, Gyeongsan City 38430, Korea; 3College of Pharmacy, Korea University, Sejong City 30019, Korea; kim_ki@korea.ac.kr

**Keywords:** chronic diseases, diabetes, dyslipidemia, hypertension, semiconductor, workplace health promotion

## Abstract

(1) Background: This study aimed to analyze the risk of chronic diseases including hypertension, diabetes, and dyslipidemia in workers of a semiconductor manufacturing company and the factors associated with their participation in workplace health promotion (WHP) programs. (2) Methods: Subjects were workers in a semiconductor and liquid crystal display company in South Korea who had undergone regular health checkups. Data from regular health checkups and WHP interventions were analyzed. Health risk was classified based on the diagnosed disease, in-house classification criteria, and pooled cardiovascular risk score. (3) Results: The baseline characteristics of 39,073 participants included the following: male, 67.8%; between 30 and 40 years of age, 74.1%; <2 h of physical activities, 65.9%. Workers at risk of chronic diseases accounted for 22.2%, and 20.1% were suspicious cases of chronic diseases. Body mass index, and cholesterol level were relatively high in workers with the burden of chronic diseases. The participation rate in WHP programs was 28.8% in a high-risk group among workers at risk of chronic diseases. More participation was shown in male, older age groups, production work type, and single-person household. (4) Conclusions: Because of the low participation rate in WHP activities among workers with the burden of chronic diseases, it is necessary to establish measures to encourage their participation.

## 1. Introduction

The semiconductor and liquid crystal display (LCD) industry has been largely driven by the demand for cutting-edge electronic devices such as desktops, laptops, and wireless communication products. Moreover, cloud-based computing and modern life greatly depend on the products of this industry. The growth of the semiconductor/LCD industry will continue with new applications for the high-performance computer market segment [1]. The high-tech industry originated in developed countries such as the United States, Japan, and Europe, and then migrated globally to find niches in Korea, Taiwan, Vietnam, China, and other Asian countries because of their low wage rates, highly skilled workers, and limited regulations pertaining to the environment and occupational health [2,3]. Since the health of workers in this type of industry is one of the key indicators of labor productivity and represents the level of human rights and welfare of the workers, personal and social interests in workplace health promotion (WHP) programs of workers in the semiconductor/LCD industries have increased [4].

The prevalence of chronic diseases such as hypertension, diabetes, and dyslipidemia among industry workers is increasing because of their irregular lifestyle and work environment. It has been reported that the implementation of WHP programs for workers with chronic diseases improves their health indicators such as blood glucose levels, blood pressure, and total cholesterol levels [5,6]. In addition, with positive outcomes of increased productivity, improved employee morale, and reduced employee turnover, the implementation of WHP programs is expanding [4,7]. According to the World Health Organization (WHO), chronic diseases account for 71% of deaths worldwide, which is also considered a major cause of death in Korea over the past decade [8,9]. Thus, the importance of preventing and managing chronic diseases in workers’ health management is drawing attention. A study reported that there are significantly more men in the high-risk group for cardiovascular diseases (CVDs) than are women among adults between 40 and 50 years of age in Korea [10]. Considering that among semiconductor workers in Korea over 40 years of age, males account for 29.4% and females 0.7% [11], it is necessary to analyze the chronic diseases status of these workers and increase the WHP participation rate.

According to the laws and regulations in relation to occupational safety and health in Korea, employers should conduct health examinations for the health management of workers, which include the General Health Examinations, or the Special Health Examinations provided by the government or employer [12,13]. Therefore, it is necessary to evaluate the health status of employees in the workplace using their health checkup data, and to find ways for the improvement of their health conditions. The aim of the present study was to analyze the risk of chronic diseases including hypertension, diabetes, and dyslipidemia, in workers of a semiconductor manufacturing company, and the factors associated with their participation in WHP programs according to different risk levels of chronic diseases.

## 2. Materials and Methods

### 2.1. Study Population

The subjects of this study were workers who had joined a semiconductor/LCD company located in the Republic of Korea before 2016 and had undergone regular health checkups such as the General Health Examinations, the Comprehensive Health Examinations, or the Special Health Examinations at a designated tertiary hospital in 2016. The General Health Examination is conducted for all employees by the government or employer to detect obesity, dyslipidemia, high blood pressure, diabetes, etc., which are risk factors for cardiovascular and cerebrovascular diseases, at an early stage. It is provided once every two years for office workers and once a year for non-office workers. The Comprehensive Health Examination provided by the employer is a type of health checkup for employees ≥30 years of age that is to improve the quality of life by early detection of cancer, and CVDs, which are the top three causes of death in Korea, during asymptomatic conditions. The Special Health Examination is provided for workers engaged in areas with exposure to hazardous factors once every six months to two years based on the type of hazards. It is conducted at a designated hospital by the Ministry of Employment and Labor and the employer [12,13]. Based on the result of the health check-ups, participants were classified into one of the following health management groups: A, healthy group; C, suspicious cases of disease; D, cases to be observed for disease; R, cases subject to a follow-up health checkup; and the U and R groups who did not attend a recommended follow-up health checkup [12]. Employees of contractor companies and those who resigned before 2016 were excluded from the analysis.

The risk of CVDs in employees was assessed using the 10-year atherosclerotic cardiovascular disease (ASCVD) risk score [14] and the Framingham risk score [15]. The 10-year ASCVD risk score was calculated for employees ≥40 years of age and the Framingham risk score was calculated for employees ≥30 years of age with the demographic, survey, and clinical information. The subjects with insufficient information for the calculation of each CVD risk score were excluded from the analysis.

### 2.2. Data Source

This study was based on the health checkup data, including the General Health Examinations, Complete Health Examinations, and Special Health Examinations, in 2016, conducted at a designated tertiary hospital for employees of a semiconductor/LCD company located in Korea. Health checkup data included demographic characteristics, survey data on medical history, medication history, and smoking history, physical examination, vital signs, laboratory results, CVD risk by the Korea Occupational Safety and Health Agency (KOSHA), health management groups as mentioned in Section 2.1, and information on workers subject to the Special Health Examinations [12,16].

The 10-year ASCVD risk scores and the Framingham risk scores were calculated using demographic characteristics, survey data, and laboratory results from three different health checkup data [14,15]. The WHP programs provided by the semiconductor/LCD company for the employees included 1:1 personal training, musculoskeletal exercise center, health lecture, and individual counseling. Participation data of WHP programs in 2016 and personnel data of the participants were provided by the company. The participation data included the type and date of the program utilized by employees of the semiconductor/LCD company. Personnel data included demographic characteristics of employees, work characteristics, and household information. Data that were missing in >40% of the subjects were excluded from the analysis [17]. This study was approved by the Institutional Review Board of Seoul National University (IRB No. 1706/002-007, 9 June 2017). All data were anonymized before analysis. Data analyses were conducted from February 2017 to June 2018.

### 2.3. Classification of Health Risk Groups

The semiconductor/LCD company had an in-house standard to evaluate the health status of employees and selected eligible populations for the WHP programs based on their health status. In this study, we applied various classification criteria focusing on chronic diseases such as hypertension, diabetes, and dyslipidemia to the employees to evaluate risks of chronic diseases and their participation in WHP programs in terms of chronic diseases.

Risk levels of chronic diseases in employees of the semiconductor/LCD company were classified into three groups according to the following criteria. If there was a history of chronic disease in the survey data of the health checkup or if the subject was classified into the D group among the health management groups as a result of the health checkup, the subject was classified as a worker at risk of chronic diseases. Those who were classified in the C group for hypertension, diabetes, or dyslipidemia as a result of the regular health checkup were classified as a suspicious case of chronic diseases. The remaining workers, excluding workers at risk of chronic diseases, and suspicious workers of chronic diseases, those in R or U groups, were classified as healthy workers.

The workers at risk of chronic diseases were further classified as low-, intermediate- and high-risk groups. We modified the company’s in-house criteria (unpublished) to classify risk groups in workers at risk of chronic diseases as follows. (1) The high-risk group comprised of those with a risk of three of the chronic diseases (hypertension, diabetes, and dyslipidemia), those in a high-risk group in the KOSHA CVD risk assessment, those with a systolic blood pressure/diastolic blood pressure (SBP/DBP) ≥160/100 mmHg, or those with a body mass index (BMI) of ≥35 kg/m^2^. (2) The intermediate-risk group comprised those with the risk of two of the considered chronic diseases, those in an intermediate-risk group according to the KOSHA CVD risk assessment, those with a SBP/DBP 140–159/90–99 mmHg, or those with a BMI of 30–34.9 kg/m^2^. (3) The low-risk group was defined as those with the risk of one of the considered chronic diseases, those in a low-risk group based on the KOSHA CVD risk assessment, those with a SBP/DBP 120–139/80–89 mmHg, or those with a BMI of 25–29.9 kg/m^2^ [16,18,19]. The original in-house criteria for the health status assessment included diabetes, hyperlipidemia, liver disease, and anemia in the category of chronic diseases; however, in this study, the target chronic diseases were hypertension, diabetes, and hyperlipidemia, considering the study’s purpose.

According to the result of the 10-year ASCVD risk score, the subjects were divided into either a low- or high-risk group. The low-risk group were those with a risk score of <7.5%, and the high-risk group were those with a risk score of ≥7.5% [14]. According to the result of the Framingham risk score, subjects were classified as low-risk for a risk score of <10%, the intermediate-risk group for a risk score of 10–20%, and the high-risk group was comprised of those with a risk score of ≥20% [15].

### 2.4. Statistical Analysis

Missing demographic and clinical data were substituted with the previous year’s data. Data were shown as numbers and percentages for categorical variables, and as means and standard deviations (SD) for continuous data. Fisher’s exact and chi-square tests were used to compare categorical data, and unpaired *t*-tests and Mann-Whitney tests were used to compare continuous data. Cramer’s V was measured to analyze the association between risk levels of chronic diseases (healthy workers, suspicious cases of chronic diseases, and workers at risk of chronic diseases) and each CVD risk score. For analysis of demographic factors influencing participation in health promotion activities, a multivariable logistic regression analysis was performed. Odds ratios (ORs) and their 95% confidence intervals (CIs) were calculated with a stepwise selection approach to simultaneously include variables in the model if they were normally associated with the occurrence of the participation in univariable analysis. Statistical significance was set at a two-sided *p*-value < 0.05, and all data analyses were conducted using SAS version 9.3 (SAS Institute, Cary, NC, USA).

## 3. Results

### 3.1. Demographic Characteristics According to the Health Risk Groups

As shown in Table 1, the total number of study participants was 39,073. A total of 97.9% of the study population were aged ≤40 years, and 67.8% were male. A total of 86.0% of all participants worked in the device solution division, 57.5% worked in the production division, and 44.5% worked in the fabrication division. A total of 65.9% of the workers responded that they spent less than two hours on physical activities at work, and 66.3% had undergone special health checkups due to exposure to harmful substances.

Regarding the distribution of risk groups according to chronic diseases based on the health checkup data of employees, 8682 workers had a risk of chronic diseases, such as hypertension, diabetes, or dyslipidemia (22.2%), 7865 workers were the suspicious cases of chronic diseases (20.1%), and 22,342 healthy workers did not have abnormal findings (57.2%). In terms of type of chronic disease, 3389 workers were at risk of hypertension (8.7%), 528 workers at risk of diabetes (1.4%), and 6379 workers at risk of dyslipidemia (16.3%). Only 9.7% of female workers had risk of chronic diseases as opposed to 28.2% of male workers. As age increased, the proportion of healthy workers decreased (55.9%, 41.2%, and 34.1% in the 30s, 40s, and 50s or older age groups, respectively) and that of workers at risk of chronic diseases increased (21.6%, 36.9%, and 48.0% in the 30s, 40s, and 50s or older age groups, respectively). When the physical activity at the workplace in hour-units was less than four hours, the proportion of workers at risk of chronic diseases was 23.2% (8409 out of 36,270 workers), compared to the 9.7% found in employees with more than four hours of physical activity. Whereas only 14.9% of workers at risk of chronic diseases were from single-person households, 28.4% were from households with 6–9 people. A greater proportion of workers who lived in a dormitory were workers at risk of chronic diseases than those who did not (23.0% vs. 9.0%, respectively). Among the workers subject to special health checkups, 40.6% of them were at risk of chronic diseases or suspicious of chronic diseases. The proportion of workers at risk of chronic diseases was comparable between subject to special health checkups (20.5%) and the other workers (25.7%). Similarly, the risk of chronic diseases did not show a substantial difference between production employees who are more likely to be exposed to electronic materials (production, 21.2%) and office workers (23.7%).

When 8682 workers (100%) at risk of chronic diseases were classified according to their risk classification criteria, 3961 workers were classified as the low-risk group (45.6%), 3792 workers as the intermediate-risk group (43.7%), and 929 workers as the high-risk group (10.7%) (Appendix A). In addition, regarding the risk of CVDs among 7447 employees at the semiconductor/LCD workplace, 95 workers (1.3%) were classified as the high-risk group with the 10-year ASCVD risk score of ≥7.5%. Based on the Framingham risk score among 26,456 employees at the semiconductor/LCD workplace, 2965 workers (11.2%) were classified as the intermediate-risk group (10 to <20%), and 486 workers (1.8%) were classified as the high-risk group with the risk score of ≥20% (Appendix A).

Table 2 shows the distribution of each risk level of chronic diseases according to the risk classification criteria in workers at risk of chronic diseases and CVD risk groups. When workers at risk of chronic diseases were classified as the high-risk group, 61.1% (568/929), 38.1% (354/929) and 63.0% (585/929) of them had a risk of hypertension, diabetes or dyslipidemia, respectively. Among 3792 workers in the intermediate-risk group, 81.8% (3102/3792) were at risk of dyslipidemia. Even in the low-risk group, 1.0% (38/3961) of them had a risk of diabetes, whereas 68.0% (2692/3961) had a risk of dyslipidemia. Among the employees in their 40s or older, a correlation coefficient between risk levels of chronic diseases (healthy workers, suspicious cases of chronic diseases, and workers at risk of chronic diseases) and CVD risk groups based on 10-year ASCVD risk score (<7.5%, and ≥7.5%) was 0.120, and 93.7% of those with an ASCVD risk score ≥7.5% (89/95) were classified as workers at risk of chronic diseases. Among the employees in their 30s or older, 86.6% of those with a Framingham risk score ≥20% (421/486) were classified as workers at risk of chronic diseases with a correlation coefficient of 0.228. The chronic diseases were suspicious in a total of 27.8% (960/3451) of participants with a Framingham risk score ≥10%.

### 3.2. Clinical Characteristics According to Risk Levels of Chronic Diseases

As shown in Table 3, the BMI, blood pressure, fasting glucose levels, and total cholesterol levels were higher in suspicious cases of chronic diseases or for workers at risk of chronic diseases compared to healthy workers. Healthy workers had a BMI of 22 ± 2.6 kg/m^2^. Suspicious cases of chronic diseases had a higher BMI of 27 ± 3.5 kg/m^2^ than those at risk of chronic diseases (26 ± 3.7 kg/m^2^). Similar SBP/DBP values were observed between the suspicious cases of the chronic diseases (120/72 mmHg) and workers at risk of chronic diseases (122/76 mmHg). Workers at risk of diabetes had fasting glucose and hemoglobin A1c (HbA1c) levels of 145 ± 52.1 mg/dL and 7.5 ± 1.7%, respectively. On the other hand, suspicious cases of chronic diseases and workers at risk of chronic diseases showed similar fasting glucose (94 ± 10.7 mg/dL and 99 ± 22.5 mg/dL, respectively) and HbA1C (5.5 ± 0.3% and 5.7 ± 0.8%, respectively) levels. Healthy workers showed mean triglyceride and LDL-C levels of 83 and 111 mg/dL, respectively. These levels were 2.5- and 1.5-fold higher in workers at risk of dyslipidemia, respectively (206 mg/dL and 157 mg/dL, respectively). Suspicious cases of the chronic diseases showed triglyceride and LDL-C levels of 142 ± 68.5 mg/dL and 134 ± 26.9 mg/dL, respectively.

### 3.3. Factors Influencing the Participation in the Workplace Health Promotion Program

Figure 1 and Appendix A show the participation rate of employees in WHP programs according to the health risk classification. According to the risk levels of chronic diseases, 407 workers at risk of chronic diseases (4.7%, 407/8982) participated in WHP activities such as health camps, personal training, musculoskeletal exercise, or 1:1 counselling in 2016. It was a significantly higher rate of participation compared to workers suspicious of chronic diseases (0.5%, 40/7865, *p* < 0.001). In terms of chronic diseases, the participation rate of workers at risk of diabetes was the highest (27.1%, 143/528), and the rate of workers at risk of dyslipidemia was the lowest (3.8%, 242/6379). Among the workers at risk of chronic diseases, the participation rate of the high-risk group was 28.8% (268/929), which was significantly higher than that of the intermediate-risk group (2.8%, 108/3792, *p* < 0.001). The group with a 10-year ASCVD risk score ≥7.5% and the group with a Framingham risk score ≥20%, all of which are the high-risk groups of CVD diseases, showed participation rates of 11.6% and 6.0%, respectively. These rates were significantly higher than those of the low-risk groups (2.2% for the group with a 10-year ASCVD risk score <7.5% and 1.2% for the group with a Framingham risk score <10%, *p* < 0.001).

Table 4 shows the factors influencing participation in WHP programs among employees working in the semiconductor/LCD company. Male workers showed a significantly higher rate of participation in WHP activities than did female workers (OR, 2.535; 95% CI, 1.915–3.413). The participation rate increased as age increased (OR (95% CI), 1.813 (1.305–2.560), 3.798 (2.590–5.636), and 3.987 (2.111–7.232) for workers in their 30s, 40s, and 50s or older with respect to those in their 20s, respectively). Production workers showed a higher participation rate than did office workers (OR, 1.503; 95% CI, 1.205–1.887). In addition, certain workplaces showed significantly low participation rates (OR (95% CI), 0.214 (0.109–1.383) for Workplace D with respect to the reference). Workers from single-person households showed a significantly higher participation rate compared to the reference (OR, 1.505; 95% CI, 1.072–2.104).

## 4. Discussion

Health is a comprehensive concept encompassing physical, mental, and social health, and health promotion comprises a series of processes for controlling and improving one’s own health. It was reported that the health of workers in the semiconductor industry were at risk due to widespread use of harmful chemical substances. Approximately 33% of the products contained trade secret ingredients, which might cause a lack of hazardous information [1]. In this study, the demographic and clinical characteristics of employees of the semiconductor/LCD company, which are associated with the risk of exposure to various types of hazards, were analyzed by health risk groups with a focus on chronic diseases. In addition, the participation rate for WHP activities at the workplace according to health risk groups and the factors influencing participation in the activities were investigated. To the best of our knowledge, this is the first cross-sectional analysis of the overall status of chronic diseases among workers in this industry with exposure to hazardous chemicals and their participation in WHP activities.

Examining the distribution of health risk groups according to diagnosed chronic diseases, 22.2% of workers in the semiconductor/LCD company had risk of chronic diseases such as hypertension, diabetes, and dyslipidemia, and 20.1% were suspicious cases of chronic diseases. The distribution of chronic disease risk did not show a substantial difference by exposure to hazardous factors or by work type such as office work or production. Exposure to short wavelength lighting emitted from computer screens may have a detrimental effect on sleep or circadian regulation [20]. This study also showed that exposure to computer screens could have a negative effect on the occurrence of chronic diseases as much as the effects of substances exposed during the semiconductor manufacturing process on the chronic diseases. The proportion of workers with a high risk of chronic diseases was higher among men and increased with age, and these results are consistent with those of the analysis on the prevalence of chronic diseases in Korean adults who were seldom exposed to harmful substances extensively [21]. In our study, the proportion of workers who were at risk of hypertension or diabetes in the semiconductor/LCD company was lower than their prevalence reported for those in their 30s–40s in the Korea National Health and Nutrition Examination Survey (8.7% vs. 27.9% and 1.4% vs. 9.5% for hypertension and diabetes, respectively) [21]. It was shown that the mean SBP/DBP was 128/80 mmHg even among the workers at risk of hypertension, indicating blood pressure was being well-managed under 140/90 mmHg, the cut-off value for hypertension in this group [18]. Fasting glucose and HbA1C levels were also maintained under 126 mg/dL and 6.5%, respectively, for all workers, except for workers at risk of diabetes [22]. However, considering that mean HbA1c level, which reflects the long-term control status of the blood glucose level, was 7.5% in workers at risk of diabetes, and their mean fasting glucose level was 145 mg/dL, it is necessary to prepare an intensive management plan for diabetes in these workers [5]. The workers at risk of dyslipidemia in this study, were 20.6% among men, which was higher than that of women (7.4%), and 97.9% of the study participants were aged <50 years. Considering that the prevalence of hypercholesterolemia was 10.8% and 20.2% among average Korean men in their 30s and 40s, respectively, the prevalence in male employees working in the semiconductor/LCD company was somewhat higher [21]. The workers at risk of chronic diseases had a mean total cholesterol level and triglyceride level of 219 mg/dL and 185 mg/dL, respectively. Workers at risk of chronic diseases and suspicious cases of chronic diseases, who accounted for 42.3% of all workers, had LDL-C levels of 149 and 134 mg/dL, respectively. According to Korean diagnostic criteria for dyslipidemia, the cholesterol levels in these workers at risk of chronic diseases were at borderline levels (200–239 mg/dL, 150–199 mg/dL, and 130–159 mg/dL for total cholesterol, triglyceride and LDL-C, respectively) [23]. However, the adequacy of dyslipidemia management could be determined in consideration of the risk of major cardiovascular event at each individual level [23], so further study on this is needed. The said groups had a mean BMI of 27 and 26 kg/m^2^, respectively, which were higher than the cut-off value for obesity [24]. Considering that 65.9% of all workers spend less than two hours on physical activities at their workplaces, it is necessary to develop an intensive nutrition and physical activity program at the company level to manage obesity among workers [25,26]. It was reported that endocrine disrupting chemicals were associated with metabolic disorder such as obesity, hyperlipidemia and type 2 diabetes [27]. Since the semiconductor workplaces deal with products with various trade secret ingredients, businesses and governments should evaluate the relationship between the working environment in the semiconductor/LCD companies and the incidence of various metabolic disorders as well as the CVDs which might be caused by these disorders.

This study showed that the proportion of workers at risk of chronic diseases was 24.8% for workers with normal working hours, whereas it was 9.4% for workers with a flexible schedule. According to a recent report from the World Health Organization (WHO) and the International Labour Organization (ILO) on the work-related burden of disease, working more than 49 h per week was significantly correlated with the increased risk of acquiring stroke compared to workers who worked less than 40 h/week [28]. In Korea, the legal working hours in Korea were 40 h per week, and was allowed up to 52 h/week in 2016, which might cause serious health problems for workers, especially in the working environments exposed to hazardous chemical substances [1,29]. Thus, it is necessary to make a systematic plan to improve the health of Korean workers through continuous research on the correlation between working hours and their health status.

The suspicious cases of chronic diseases accounted for 20.1% of all workers, and 14.9% and 22.1% of workers in their 20s and 30s, who are generally at low risk of chronic diseases, were in this group [21]. It was reported that systematic, intensive lifestyle interventions, such as chronic disease prevention program at the workplace, were effective in reducing risk factors for chronic diseases [30,31]. Therefore, it is necessary to develop and regularly assess a WHP program tailored to workers suspicious of chronic diseases based on their demographic and clinical characteristics.

Unlike in-house health risk classification criteria that comprehensively consider the number of chronic diseases, the KOSHA CVD risk, blood pressure, and BMI, the risk levels of chronic disease used in this study considered not only health check-up data, but also the medical history from survey data from off-site medical facilities. Among the workers at risk of chronic diseases, 45.6% and 43.7% of the workers were classified as the low-risk and intermediate-risk groups, respectively. However, most of the workers at risk of hypertension and dyslipidemia were in the low-risk or intermediate-risk groups (83.2%, 2821/3389 and 90.8%, 5794/6379, respectively). Considering the workers’ obesity and cholesterol levels presented in this study, it is necessary to review the in-house criteria of health risk with a focus on chronic diseases. Workers at risk of diabetes were mostly distributed in the high-risk group (67.0%, 354/528), which might mean that more attention is needed regarding the health risk of workers at risk of diabetes.

Since hypertension, diabetes and dyslipidemia are associated with the occurrence of CVDs significantly, we analyzed the risk of CVDs in the employees using the Framingham and ASCVD risk scores [32]. The Framingham risk score was more strongly correlated with the risk levels of chronic diseases used in this study compared to the 10-year ASCVD risk score. Although it was reported that these pooled cohort equations were not appropriate to estimate risk of the ASCVD in Korea, both the KOSHA CVD risk assessment and the Framingham risk scores might be considered to predict and manage the risk of CVDs in these workers in the absence of a validated CVD prediction model for Koreans [33,34]. Approaches to intensively manage workers’ health by comprehensively considering various health risk classification systems is necessary to prevent and efficiently manage common chronic diseases.

When we analyzed the participation rate of WHP activities considering chronic diseases such as hypertension, diabetes, and dyslipidemia, the participation rate was 28.8% annually among those with a high risk for chronic diseases. This value is in the range of 10–64% (median (95% CI), 33% (25–42%)) reported in the literature for participation rates in WHP activities [35]. Focusing on all study populations in the company, the participation rate became 4.7% annually. Workers suspicious of chronic diseases had an especially low participation rate of <1%, despite them being at some risk of chronic diseases considering their obesity and high cholesterol levels. Male workers, older workers, workers from a single-person household, and production workers participated relatively more actively in the WHP activities. These results are similar to those of a previous study [35]. Sorensen et al. reported that it is necessary to focus on individual health-related behaviors, the work environment, and the work, family, and community interface as the intervention target for worksite chronic disease prevention [36]. Since WHP programs involve a relatively homogeneous population group in the workplace and could ensure the participation of a constant number of people, they can facilitate continuous health management. It is necessary to devise various methods, such as providing incentives, education programs, fees, etc., to increase workers’ participation in WHP activities after considering factors such as the workers’ environment, age, sex, and demographic/lifestyle characteristics. The short- and long-term effects of these methods must also be analyzed.

This study has some limitations that need to be considered. First, this study focused on a single company in Korea. It was reported that there were 1224 semiconductor companies in Korea, employing 16,794 people, of which 84,751 (50.5%) were working in large corporations in 2018 [37]. In this study, we analyzed the data of a total of 39,073 employees from a semiconductor company, which accounted for 46.1% of all employees of large semiconductor companies in Korea. Therefore, the results of this study might be representative of the whole working population in the Korean semiconductor industry. Second, due to its cross-sectional design, this study did not investigate long-term trends in chronic diseases and participation in WHP activities. However, it is a large-scale study that included 39,073 workers who had been working at a semiconductor industry until 2016, and sufficiently examined the recent situation regarding chronic diseases among them and their participation in WHP activities. Lastly, this study did not investigate the effect of WHP activities among suspicious workers of chronic diseases and those at risk of chronic diseases. Additional research is needed to examine the effect of such interventions.

## 5. Conclusions

In conclusion, a total of 42.3% of employees of a semiconductor/LCD company were classified as workers at risk of chronic diseases or suspicious cases of chronic diseases, indicating concerns for notable health risks. Because of the low participation rate in WHP activities among them, it is necessary to establish measures to encourage their participation.

## Figures and Tables

**Figure 1 ijerph-18-11383-f001:**
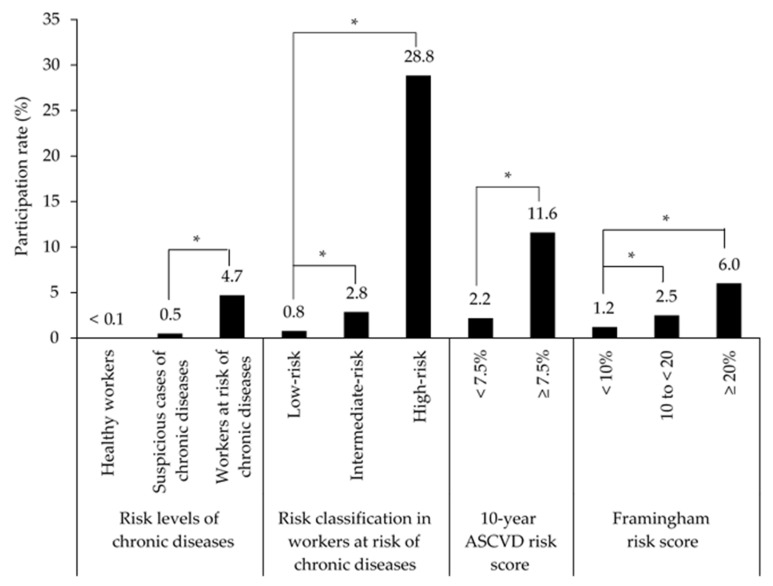
Participation rates for health promotion programs in health risk groups among employees of the semiconductor/liquid crystal display (LCD) company. ASCVD, atherosclerotic cardiovascular disease. * *p* < 0.001.

**Table 1 ijerph-18-11383-t001:** Demographic characteristics of employees in the semiconductor/liquid crystal display (LCD) workplace with regard to risk levels of chronic diseases.

Characteristics		Total, *n* (%)		Healthy Workers, *n* (%)		Suspicious Cases of Chronic Diseases, *n* (%)		Workers at Risk of Chronic Diseases, *n* (%)							
								Chronic Diseases ^a^		Hypertension		Diabetes		Dyslipidemia	
Total		39,073	(100.0)	22,342	(57.2)	7865	(20.1)	8682	(22.2)	3389	(8.7)	528	(1.4)	6379	(16.3)
General characteristics	Sex														
	Male	26,481	(67.8)	12,548	(47.4)	6302	(23.8)	7464	(28.2)	3089	(11.7)	440	(1.7)	5452	(20.6)
	Female	12,592	(32.2)	9794	(77.8)	1563	(12.4)	1218	(9.7)	300	(2.4)	88	(0.7)	927	(7.4)
	Age groups														
	20s or younger	9262	(23.7)	7044	(76.1)	1384	(14.9)	802	(8.7)	275	(3.0)	45	(0.5)	553	(6.0
	30s	20,918	(53.5)	11,694	(55.9)	4618	(22.1)	4509	(21.6)	1732	(8.3)	180	(0.9)	3293	(15.7)
	40s	8057	(20.6)	3319	(41.2)	1714	(21.3)	2970	(36.9)	1193	(14.8)	242	(3.0)	2244	(27.9)
	50s or older	836	(2.1)	285	(34.1)	149	(17.8)	401	(48.0)	189	(22.6)	61	(7.3)	289	(34.6)
Work characteristics	Division														
	Device solution	33,609	(86.0)	19,135	(56.9)	6751	(20.1)	7563	(22.5)	2919	(8.7)	459	(1.4)	5610	(16.7)
	LCD	5464	(14.0)	3207	(58.7)	1114	(20.4)	1119	(20.5)	470	(8.6)	69	(1.3)	769	(14.1)
	Work type														
	Office work	16,605	(42.5)	9226	(55.6)	3363	(20.3)	3928	(23.7)	1492	(9.0)	234	(1.4)	2916	(17.6)
	Production	22,468	(57.5)	13,116	(58.4)	4502	(20.0)	4754	(21.2)	1897	(8.4)	294	(1.3)	3463	(15.4)
	Industry														
	Fabrication	17,381	(44.5)	10,358	(59.6)	3406	(19.6)	3544	(20.4)	1424	(8.2)	228	(1.3)	2562	(14.7)
	Support	2968	(7.6)	1699	(57.2)	559	(18.8)	698	(23.5)	299	(10.1)	54	(1.8)	494	(16.6)
	Development	18,553	(47.5)	10,150	(54.7)	3880	(20.9)	4424	(23.8)	1662	(9.0)	245	(1.3)	3308	(17.8)
	Part-time	171	(0.4)	135	(78.9)	20	(11.7)	16	(9.4)	4	(2.3)	1	(0.6)	15	(8.8)
	Type of work shift														
	Normal working hours	28,549	(73.1)	15,386	(53.9)	5943	(20.8)	7082	(24.8)	2794	(9.8)	441	(1.5)	5227	(18.3)
	4 sets of 3 shifts	7309	(18.7)	4985	(68.2)	1250	(17.1)	1040	(14.2)	358	(4.9)	56	(0.8)	756	(10.3)
	Variable shift	3044	(7.8)	1836	(60.3)	652	(21.4)	544	(17.9)	233	(7.7)	30	(1.0)	381	(12.5)
	Flexible schedule	171	(0.4)	135	(78.9)	20	(11.7)	16	(9.4)	4	(2.3)	1	(0.6)	15	(8.8)
	Physical activities at the workplace (hours)														
	<2	25,740	(65.9)	14,885	(57.8)	5071	(19.7)	5667	(22.0)	2101	(8.2)	344	(1.3)	4215	(16.4)
	2 to <4	10,530	(26.9)	5357	(50.9)	2369	(22.5)	2742	(26.0)	1217	(11.6)	169	(1.6)	1959	(18.6)
	4 to <6	2803	(7.2)	2100	(74.9)	425	(15.2)	273	(9.7)	71	(2.5)	15	(0.5)	205	(7.3)
	Workplace ^b^														
	A	9012	(23.1)	5203	(57.7)	1762	(19.6)	2019	(22.4)	834	(9.3)	138	(1.5)	1473	(16.3)
	B	14,943	(38.2)	8407	(56.3)	3032	(20.3)	3446	(23.1)	1304	(8.7)	197	(1.3)	2609	(17.5)
	C	5789	(14.8)	3186	(55.0)	1253	(21.6)	1310	(22.6)	434	(7.5)	60	(1.0)	1004	(17.3)
	D	4115	(10.5)	2474	(60.1)	748	(18.2)	857	(20.8)	381	(9.3)	69	(1.7)	569	(13.8)
	E	5214	(13.3)	3072	(58.9)	1070	(20.5)	1050	(20.1)	436	(8.4)	64	(1.2)	724	(13.9)
	Duration employed (years)														
	≤5	8234	(21.1)	5584	(67.8)	1603	(19.5)	998	(12.1)	348	(4.2)	38	(0.5)	718	(8.7)
	6–10	12,287	(31.4)	7467	(60.8)	2500	(20.3)	2276	(18.5)	887	(7.2)	99	(0.8)	1624	(13.2)
	11–20	14,559	(37.3)	7670	(52.7)	3017	(20.7)	3806	(26.1)	1432	(9.8)	211	(1.4)	2854	(19.6)
	>20	3993	(10.2)	1621	(40.6)	745	(18.7)	1602	(40.1)	722	(18.1)	180	(4.5)	1183	(29.6)
Household information	Number of household members														
	1	5448	(13.9)	3660	(67.2)	952	(17.5)	814	(14.9)	307	(5.6)	46	(0.8)	592	(10.9)
	2	5962	(15.3)	3779	(63.4)	1131	(19.0)	1018	(17.1)	370	(6.2)	46	(0.8)	756	(12.7)
	3–5	25,517	(65.3)	13,826	(54.2)	5333	(20.9)	6241	(24.5)	2469	(9.7)	398	(1.6)	4576	(17.9)
	≥6	2146	(5.5)	1077	(50.2)	449	(20.9)	609	(28.4)	243	(11.3)	38	(1.8)	455	(21.2)
	Household types														
	Single-person household	5448	(13.9)	3660	(67.2)	952	(17.5)	814	(14.9)	307	(5.6)	46	(0.8)	592	(10.9)
	Couple	4086	(10.5)	2458	(60.2)	816	(20.0)	784	(19.2)	266	(6.5)	24	(0.6)	608	(14.9)
	Couple + children	13,617	(34.9)	6356	(46.7)	3084	(22.6)	4113	(30.2)	1623	(11.9)	283	(2.1)	3072	(22.6)
	Couple + parents	1999	(5.1)	1228	(61.4)	417	(20.9)	350	(17.5)	122	(6.1)	13	(0.7)	258	(12.9)
	Couple + children + parents	3982	(10.2)	1871	(47.0)	903	(22.7)	1186	(29.8)	465	(11.7)	79	(2.0)	888	(22.3)
	Others	9941	(25.4)	6769	(68.1)	1693	(17.0)	1435	(14.4)	606	(6.1)	83	(0.8)	961	(9.7)
	Dormitory														
	Yes	36,775	(94.1)	20,636	(56.1)	7492	(20.4)	8476	(23.0)	3298	(9.0)	514	(1.4)	6249	(17.0)
	No	2298	(5.9)	1706	(74.2)	373	(16.2)	206	(9.0)	91	(4.0)	14	(0.6)	130	(5.7)
Exposure to hazardous factors	Special health checkups														
	Not applicable	13,184	(33.7)	7072	(53.6)	2650	(20.1)	3387	(25.7)	1285	(9.7)	227	(1.7)	2537	(19.2)
	Subject of examination	25,889	(66.3)	15,270	(59.0)	5215	(20.1)	5295	(20.5)	2104	(8.1)	301	(1.2)	3842	(14.8)

^a^ Hypertension, diabetes, or dyslipidemia. ^b^ Main process of workplaces: A and B, wafer fabrication; C, research and development; D and E, test and packaging.

**Table 2 ijerph-18-11383-t002:** Correlation among health risk classifications in employees of the semiconductor/liquid crystal display (LCD) company.

Criteria	Risk Classification	Total, *n* (% ^a^)		Risk Levels of Chronic Diseases											
				Healthy Workers, *n* (%)		Suspicious Cases of Chronic Diseases, *n* (%)		Workers at Risk of Chronic Diseases, *n* (%)							
								Chronic Diseases ^b^		Hypertension		Diabetes		Dyslipidemia	
Risk classification in workers at risk of chronic diseases	Low-risk group	3961	(10.1)	0	NA	0	NA	3961	(100.0)	1231	(31.1)	38	(1.0)	2692	(68.0)
	Intermediate-risk group	3792	(9.7)	0	NA	0	NA	3792	(100.0)	1590	(41.9)	136	(3.6)	3102	(81.8)
	High-risk group	929	(2.4)	0	NA	0	NA	929	(100.0)	568	(61.1)	354	(38.1)	585	(63.0)
10-year ASCVD risk score ^c^	<7.5%	7352	(18.8)	2483	(33.8)	1816	(24.7)	2998	(40.8)	1209	(16.4)	252	(3.4)	2274	(30.9)
	≥7.5%	95	(0.2)	2	(2.1)	4	(4.2)	89	(93.7)	48	(50.5)	24	(25.3)	71	(74.7)
Framingham risk score ^d^	<10%	23,005	(58.9)	12,121	(52.7)	5514	(24.0)	5234	(22.8)	2211	(9.6)	358	(1.6)	3647	(15.9)
	10 to <20%	2965	(7.6)	422	(14.2)	897	(30.3)	1630	(55.0)	583	(19.7)	75	(2.5)	1346	(45.4)
	≥20%	486	(1.2)	2	(0.4)	63	(13.0)	421	(86.6)	111	(22.8)	14	(2.9)	403	(82.9)

ASCVD, atherosclerotic cardiovascular disease, NA, not applicable. ^a^ Proportion of total study participants (*n* = 39,073). ^b^ Hypertension, diabetes, or dyslipidemia. ^c^ Cramer’s V coefficient between risk groups for chronic diseases and health risk groups based on 10-year ASCVD risk score = 0.120. ^d^ Cramer’s V coefficient between risk groups for chronic diseases and health risk groups based on the Framingham risk score = 0.228.

**Table 3 ijerph-18-11383-t003:** Clinical characteristics of employees of the semiconductor/liquid crystal display (LCD) company according to risk levels of chronic diseases (mean ± SD).

Clinical Characteristics	Total	Healthy Workers	Suspicious Cases of Chronic Diseases	Workers at Risk of Chronic Diseases			
				Chronic Diseases ^a^	Hypertension	Diabetes	Dyslipidemia
BMI (kg/m^2^)	24 ± 3.7	22 ± 2.6	27 ± 3.5	26 ± 3.7	27 ± 4.0	27 ± 4.2	26 ± 3.6
SBP (mmHg)	117 ± 12.6	114 ± 11.4	120 ± 11.8	122 ± 13.6	128 ± 13.8	121 ± 13.3	120 ± 12.9
DBP (mmHg)	70 ± 10.0	67 ± 8.5	72 ± 9.2	76 ± 10.6	80 ± 10.7	76 ± 10.0	74 ± 10.1
Fasting glucose (mg/dL)	92 ± 14.0	88 ± 7.8	94 ± 10.7	99 ± 22.5	100 ± 21.2	145 ± 52.1	98 ± 21.6
HbA1c (%)	5.5 ± 0.5	5.4 ± 0.2	5.5 ± 0.3	5.7 ± 0.8	5.7 ± 0.8	7.5 ± 1.7	5.7 ± 0.7
Total cholesterol (mg/dL)	194 ± 34.1	181 ± 25.0	199 ± 29.0	219 ± 41.6	203 ± 37.0	185 ± 44.1	229 ± 41.6
Triglyceride (mg/dL)	119 ± 89.6	83 ± 35.1	142 ± 68.5	185 ± 139.6	161 ± 112.1	182 ± 162.1	206 ± 153.1
HDL-C (mg/dL)	59 ± 15.9	64 ± 14.8	51 ± 13.1	52 ± 15.4	52 ± 13.7	49 ± 12.9	51 ± 15.8
LDL-C (mg/dL)	125 ± 32.9	111 ± 24.6	134 ± 26.9	149 ± 38	136 ± 34.1	118 ± 38.3	157 ± 38.4

BMI, body mass index; SBP, systolic blood pressure; DBP, diastolic blood pressure; HbA1c, hemoglobin A1c; HDL-C, high-density lipoprotein-cholesterol; LDL-C, low-density lipoprotein-cholesterol. ^a^ Hypertension, diabetes, or dyslipidemia.

**Table 4 ijerph-18-11383-t004:** Factors influencing participation in workplace health promotion programs by employees of the semiconductor/liquid crystal display (LCD) company.

Independent Variable	Number of Participants Per Total Number of Workers in the Group (%)	Adjusted Odds Ratio of the Participation (95% Confidence Interval)	*p*-Value
Sex			
Female	60/12,592 (0.5)	Reference	
Male	397/26,481 (1.5)	2.535 (1.915–3.413)	<0.001
Age group			
20s or younger	49/9262 (0.5)	Reference	
30s	224/20,918 (1.1)	1.813 (1.305–2.560)	<0.001
40s	167/8057 (2.1)	3.798 (2.590–5.636)	<0.001
50s or older	17/836 (2.0)	3.987 (2.111–7.232)	<0.001
Work type			
Office work	180/16,605 (1.1)	Reference	
Production	277/22,468 (1.2)	1.503 (1.205–1.887)	<0.001
Workplace a			
A	122/9012 (1.4)	0.893 (0.658–1.221)	0.470
B	194/14,943 (1.3)	0.838 (0.631–1.126)	0.231
C	62/5789 (1.1)	0.730 (0.493–1.082)	0.116
D	12/4115 (0.3)	0.214 (0.109–0.383)	<0.001
E	67/5214 (1.3)	Reference	
Household type			
Single-person household	63/5448 (1.2)	1.505 (1.072–2.104)	0.017
Couple	48/4086 (1.2)	1.224 (0.838–1.772)	0.289
Couple + children	175/13,617 (1.3)	0.843 (0.622–1.151)	0.277
Couple + parents	19/1999 (1.0)	0.995 (0.580–1.624)	0.983
Couple + children + parents	71/3982 (1.8)	1.226 (0.860–1.748)	0.259
Others	81/9941 (0.8)	Reference	

OR, odds ratio; CI, confidence interval. ^a^ Main process of workplaces: A and B, wafer fabrication; C, research and development; D and E, test and packaging.

## Data Availability

Not applicable.

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
