# Peer review of "Factors Influencing Workplace Health Promotion Interventions for Workers in the Semiconductor Industry According to Risk Levels of Chronic Disease"

_ijerph, 2021, doi:10.3390/ijerph182111383_

Round 1

Reviewer 1 Report

1) Please check and correct the following :

2 page 57-60 line :

“Considering that men account for 29.4% of semiconductor workers over 40s in Korea (as opposed to women accounting for only 0.7%), it is necessary to analyze the current situation regarding chronic diseases among these workers and increase their participation in WHP programs [11].”

--> Considering that among semiconductor workers in Korea over 40 years of age, males account for 29.4% and females 0.7% [11], it is necessary to analyze the chronic disease status of these workers and increase the WHP participation rate.

4 page 194 line : “23.2%” --> ??

5 page 201 line : “8,682 workers” --> 8,682 workers (100%)

5 page 204 line : “among employees” --> among 7447 employees

5 page 206 line : “Based on the Framingham risk score”

--> Based on the Framingham risk score among 26,456 employees at the semiconductor/LCD workplace,

9 page 256 line : “407 workers at risk of chronic diseases (4.7%)”

--> 407 workers at risk of chronic diseases (4.7%, 407/8,982)

2) Please include a description of the working environment (A, B, C, D, E) at the bottom of Tables 1 and 4.

3) Please reconfirm and describe the contents related to Reference [1] below.

11 page 291 line and 306 line,

4) Please reconfirm and describe the contents related to Reference [18] below.

11 page 302 line and 306 line,

12 page 316 line and 332 line

Author Response

1.       Please check and correct the following :
1.1.    2 page 57-60 line: “Considering that men account for 29.4% of semiconductor workers over 40s in Korea (as opposed to women accounting for only 0.7%), it is necessary to analyze the current situation regarding chronic diseases among these workers and increase their participation in WHP programs [11].” -> Considering that among semiconductor workers in Korea over 40 years of age, males account for 29.4% and females 0.7% [11], it is necessary to analyze the chronic disease status of these workers and increase the WHP participation rate.

According to the reviewer’s suggestion, we revised the sentence in this manuscript (page 2, lines 61-63).

1.2.       4 page 194 line : “23.2%” -> ??
According to the reviewer’s comment, we presented a detailed description of the number in the manuscript as follows: the proportion of workers at risk of chronic diseases was 23.2%, -> the proportion of workers at risk of chronic diseases was 23.2% (8,409 out of 36,270 workers) (page 5, line 200).

1.3.       5 page 201 line : “8,682 workers” --> 8,682 workers (100%)

According to the reviewer’s suggestion, we revised the sentence in this manuscript (page 5, line 207).

1.4.       5 page 204 line : “among employees” --> among 7447 employees

According to the reviewer’s suggestion, we revised the sentence in this manuscript (page 5, line 210).

1.5.       5 page 206 line : “Based on the Framingham risk score” -> Based on the Framingham risk score among 26,456 employees at the semiconductor/LCD workplace,

According to the reviewer’s suggestion, we revised the sentence in this manuscript (page 5, lines 212-213).

1.6.       9 page 256 line : “407 workers at risk of chronic diseases (4.7%)” -> 407 workers at risk of chronic diseases (4.7%, 407/8,982)

According to the reviewer’s suggestion, we revised the sentence in this manuscript (page 9, lines 270).

2.       Please include a description of the working environment (A, B, C, D, E) at the bottom of Tables 1 and 4.

According to the reviewer’s comment, we added the following description at the bottom of Table 1 and 4:
Table 1. … b Main process of workplaces: A and, B, wafer fabrication; C, research and development; D and E, test and packaging.
Table 4. … a Main process of workplaces: A and, B, wafer fabrication; C, research and development; D and E, test and packaging.

3.       Please reconfirm and describe the contents related to Reference [1] below. 11 page 291 line and 306 line,

Thank you for the reviewer’s comments. We revised the discussion section of the manuscript as follows: Health is a comprehensive concept encompassing physical, mental, and social health, and health promotion comprises a series of processes for controlling and improving one’s own health. In this study, the demographic and clinical characteristics of employees of the semiconductor/LCD company, which is associated with the risk of exposure to various types of hazards, were analyzed by health risk groups with a focus on chronic diseases [1]. -> Health is a comprehensive concept encompassing physical, mental, and social health, and health promotion comprises a series of processes for controlling and improving one’s own health. It was reported that the health of workers in the semiconductor industry was at risk due to the widespread use of harmful chemical substances. Approximately 33% of the products contained trade secret ingredients, which might cause a lack of hazardous information [1]. In this study, the demographic and clinical characteristics of employees of the semiconductor/LCD company, which is associated with the risk of exposure to various types of hazards, were analyzed by health risk groups with a focus on chronic diseases. (page 11, lines 304-309)

4.       Please reconfirm and describe the contents related to Reference [18] below. 11 page 302 line and 306 line, 12 page 316 line and 332 line

According to the reviewer’s comment, we revised the manuscript as follows: The proportion of workers with the high risk of chronic diseases was higher among men and increased with age. Considering that 97.9% of the study participants were aged < 50 years, these results are consistent with those of the analysis on the prevalence of chronic diseases in different age and sex groups in Korea [18]. In our study, the proportion of workers who had been at risk of hypertension or diabetes in the semiconductor/LCD company was lower than their prevalence reported for those in their 30’s-40’s in the Korea National Health and Nutrition Examination Survey (8.7% vs. 27.9% and 1.4% vs. 9.5% for hypertension and diabetes, respectively) [18]. It was shown that the mean SBP/DBP was 128/80 mmHg even among the workers at risk of hypertension, indicating blood pressure was being well-managed under 140/90 mmHg, the cut-off value for hypertension in this group [19]. Fasting glucose and HbA1C levels were also maintained under 126 mg/dL and 6.5%, respectively, for all workers, except for workers at risk of diabetes [20]. However, considering that mean HbA1c level, which reflects the long-term control status of the blood glucose level, was 7.5% in workers at risk of diabetes, and their mean fasting glucose level was 145 mg/dL, it is necessary to prepare an intensive management plan for diabetes in these workers [5]. Workers at risk of dyslipidemia accounted for 16.3% of the study participants, which was close to the prevalence of dyslipidemia reported for the general population in their 30’s–40’s (16.3%) [18]. -> The proportion of workers with the high risk of chronic diseases was higher among men and increased with age, and these results are consistent with those of the analysis on the prevalence of chronic diseases in Korean adults who were seldom exposed to harmful substances extensively [20]. In our study, the proportion of workers who had been at risk of hypertension or diabetes in the semiconductor/LCD company was lower than their prevalence reported for those in their 30’s-40’s in the Korea National Health and Nutrition Examination Survey (8.7% vs. 27.9% and 1.4% vs. 9.5% for hypertension and diabetes, respectively) [20]. It was shown that the mean SBP/DBP was 128/80 mmHg even among the workers at risk of hypertension, indicating blood pressure was being well-managed under 140/90 mmHg, the cut-off value for hypertension in this group [18]. Fasting glucose and HbA1C levels were also maintained under 126 mg/dL and 6.5%, respectively, for all workers, except for workers at risk of diabetes [21]. However, considering that mean HbA1c level, which reflects the long-term control status of the blood glucose level, was 7.5% in workers at risk of diabetes, and their mean fasting glucose level was 145 mg/dL, it is necessary to prepare an intensive management plan for diabetes in these workers [5]. The workers at risk of dyslipidemia in this study were 20.6% among men, which was higher than that of women (7.4%), and 97.9% of the study participants were aged < 50 years. Considering that the prevalence of hypercholesterolemia was 10.8% and 20.2% among average Korean men in their 30’s and 40’s, respectively, the prevalence in male employees working in the semiconductor/LCD company was somewhat higher [20]. (page 11-12, lines 318-338)

Reviewer 2 Report

The authors would analyze the risk of chronic diseases in workers of a semiconductor manufacturing company, and the association with their participation in WHP programs. The manuscript is well written and the study design is correct. I have some following comments, and I suggest publication after minor revision.

1. Please clarify if your sample may be representative, compared to the whole working population in Korean semiconductor industry and discuss this issue.

2. Since recent WHO-ILO studies (e.g. Deschata et al. 2020 https://doi.org/10.1016/j.envint.2020.105746) demonstrated strict correlation between long working hours (defined as working hours exceeding standard working hours, i.e. working for ≥41 h/week) and hearth diseases and stroke, please analyse your data for participants who work for ≥41 h/week and comment the related figures.

Author Response

Point 1: Please clarify if your sample may be representative, compared to the whole working population in Korean semiconductor industry and discuss this issue.

Response 1: According to the reviewer’s comments, we revised the manuscript as follows: First, this study focused on a single company in Korea. However, the worker population at this company is representative of the total worker population in the semiconductor industry, since the company has been ranked as the second largest semiconductor company in the world and the largest semiconductor company in Korea [31]. à First, this study focused on a single company in Korea. It was reported that there were 1,224 semiconductor companies in Korea, employing 16,794 people, of which 84,751 (50.5%) were working in large corporations in 2018 [36]. In this study, we analyzed the data of a total of 39,073 employees from a semiconductor company, which accounted for 46.1% of all employees of large semiconductor companies in Korea. Therefore, the results of this study might be representative of the whole working population in the Korean semiconductor industry. (page 13, lines 417-423)

Point 2: Since recent WHO-ILO studies (e.g. Deschata et al. 2020 https://doi.org/10.1016/j.envint.2020.105746) demonstrated strict correlation between long working hours (defined as working hours exceeding standard working hours, i.e. working for ≥41 h/week) and hearth diseases and stroke, please analyse your data for participants who work for ≥41 h/week and comment the related figures

Response 2: According to the reviewer’s suggestion, we revised the discussion section of the manuscript as follows: (None) à This study showed that the proportion of workers at risk of chronic diseases was 24.8% for workers with normal working hours, whereas 9.4% for workers with a flexible schedule. According to a recent report from the World Health Organization (WHO) and the International Labour Organization (ILO) on the work-related burden of disease, working more than 49 hours per week was significantly correlated with the increased risk of acquiring stroke compared to workers who worked less than 40 hours/week [26]. In Korea, the legal working hours in Korea were 40 hours per week, and were allowed up to 52 hours/week in 2016, which might cause serious health problems for workers especially in the working environments exposed to hazardous chemical substances [1,27]. Thus, it is necessary to make a systematic plan to improve the health of Korean workers through continuous research on the correlation between working hours and their health status. (page 12, lines 357-367)

Reviewer 3 Report

Song and colleagues performed a study on workers of an electronic manufacture. Participants were recruited as part of workplace health promotion (WHP) programs in south Korea to assess risk of chronic diseases. This is a well-written and easy to read article that provides insight into the risk of chronic diseases among specific group of workers in south Korea.

Major comments

In the current format the study does not produce much of a knowledge that could add value to what we already know about risk of chronic diseases. However, this could be considered as an important study if the authors would have compared those workers exposed to novel exposures vs. those who don't. Most of modern electronic devices have been added to the population exposures only in the last few decades and population studies to identify risks associated with such exposures are seldom. This is a unique sample to study such risks and the authors could use this as an opportunity to compare the risks between workers exposed to electronic material compared with those who are less exposed. I recommend adding a section to the paper to evaluate and estimate this.

Minor comments:

The authors aim seem to be evaluation of several chronic diseases. However, much of the text of the manuscript is focused on cardiovascular disease risk. 

It is not clear what reference was used to classify risk groups. For example, on what basis (SBP/DBP) ≥ 160/100 mmHg is considered as high-risk group?

In Table 1, it hasn't been clarified what characteristics A, B, C, D, and E represent. some of the subheadings are misplaced in the middle of the table.

Author Response

Point 1: In the current format the study does not produce much of a knowledge that could add value to what we already know about risk of chronic diseases. However, this could be considered as an important study if the authors would have compared those workers exposed to novel exposures vs. those who don't. Most of modern electronic devices have been added to the population exposures only in the last few decades and population studies to identify risks associated with such exposures are seldom. This is a unique sample to study such risks and the authors could use this as an opportunity to compare the risks between workers exposed to electronic material compared with those who are less exposed. I recommend adding a section to the paper to evaluate and estimate this.

Response 1: According to the reviewer’s suggestion, we revised the discussion section of the manuscript as follows: The proportion of workers with the high risk of chronic diseases was higher among men and increased with age. Considering that 97.9% of the study participants were aged < 50 years, these results are consistent with those of the analysis on the prevalence of chronic diseases in different age and sex groups in Korea [18]. In our study, the proportion of workers who had been at risk of hypertension or diabetes in the semiconductor/LCD company was lower than their prevalence reported for those in their 30’s-40’s in the Korea National Health and Nutrition Examination Survey (8.7% vs. 27.9% and 1.4% vs. 9.5% for hypertension and diabetes, respectively) [18]. It was shown that the mean SBP/DBP was 128/80 mmHg even among the workers at risk of hypertension, indicating blood pressure was being well-managed under 140/90 mmHg, the cut-off value for hypertension in this group [19]. Fasting glucose and HbA1C levels were also maintained under 126 mg/dL and 6.5%, respectively, for all workers, except for workers at risk of diabetes [20]. However, considering that mean HbA1c level, which reflects the long-term control status of the blood glucose level, was 7.5% in workers at risk of diabetes, and their mean fasting glucose level was 145 mg/dL, it is necessary to prepare an intensive management plan for diabetes in these workers [5]. Workers at risk of dyslipidemia accounted for 16.3% of the study participants, which was close to the prevalence of dyslipidemia reported for the general population in their 30’s–40’s (16.3%) [18]. The workers at risk of chronic diseases had a mean total cholesterol level and triglyceride level of 219 mg/dL and 185 mg/dL, respectively. Workers at risk of chronic diseases and suspicious cases of chronic diseases, who accounted for 42.3% of all workers, had LDL-C levels of 149 and 134 mg/dL. According to Korean diagnostic criteria for dyslipidemia, the cholesterol levels in these workers at risk of chronic diseases were at borderline levels (200–239 mg/dL, 150–199 mg/dL, and 130–159 mg/dL for total cholesterol, triglyceride and LDL-C, respectively) [21]. However, the adequacy of dyslipidemia management could be determined in consideration of the risk of major cardiovascular event at each individual level [21]. Therefore, further study on this is needed. The said groups had a mean BMI of 27 and 26 kg/m2, respectively, which were higher than the cut-off value for obesity [22]. Considering that 65.9% of all workers spend less than two hours on physical activities at their workplaces, it is necessary to develop an intensive nutrition and physical activity program at the company level to man-age obesity among workers [23,24]. -> The proportion of workers with the high risk of chronic diseases was higher among men and increased with age, and these results are consistent with those of the analysis on the prevalence of chronic diseases in Korean adults who were seldom exposed to harmful substances extensively [20]. In our study, the proportion of workers who had been at risk of hypertension or diabetes in the semiconductor/LCD company was lower than their prevalence reported for those in their 30’s–40’s in the Korea National Health and Nutrition Examination Survey (8.7% vs. 27.9% and 1.4% vs. 9.5% for hypertension and diabetes, respectively) [20]. It was shown that the mean SBP/DBP was 128/80 mmHg even among the workers at risk of hypertension, indicating blood pressure was being well-managed under 140/90 mmHg, the cut-off value for hypertension in this group [18]. Fasting glucose and HbA1C levels were also maintained under 126 mg/dL and 6.5%, respectively, for all workers, except for workers at risk of diabetes [21]. However, considering that mean HbA1c level, which reflects the long-term control status of the blood glucose level, was 7.5% in workers at risk of diabetes, and their mean fasting glucose level was 145 mg/dL, it is necessary to prepare an intensive management plan for diabetes in these workers [5]. The workers at risk of dyslipidemia in this study, were 20.6% among men, which was higher than that of women (7.4%), and 97.9% of the study participants were aged < 50 years. Considering that the prevalence of hypercholesterolemia was 10.8% and 20.2% among average Korean men in their 30’s and 40’s, respectively, the prevalence in male employees working in the semiconductor/LCD company was somewhat higher [20]. The workers at risk of chronic diseases had a mean total cholesterol level and triglyceride level of 219 mg/dL and 185 mg/dL, respectively. Workers at risk of chronic diseases and suspicious cases of chronic diseases, who accounted for 42.3% of all workers, had LDL-C levels of 149 and 134 mg/dL. According to Korean diagnostic criteria for dyslipidemia, the cholesterol levels in these workers at risk of chronic diseases were at borderline levels (200–239 mg/dL, 150–199 mg/dL, and 130–159 mg/dL for total cholesterol, triglyceride and LDL-C, respectively) [22]. However, the adequacy of dyslipidemia management could be determined in consideration of the risk of major cardiovascular event at each individual level [22], so further study on this is needed. The said groups had a mean BMI of 27 and 26 kg/m2, respectively, which were higher than the cut-off value for obesity [23]. Considering that 65.9% of all workers spend less than two hours on physical activities at their workplaces, it is necessary to develop an intensive nutrition and physical activity program at the company level to manage obesity among workers [24,25]. It was reported that endocrine-disrupting chemicals were associated with metabolic disorders such as obesity, hyperlipidemia, and type 2 diabetes [26]. Since the semiconductor workplaces deal with products with various trade secret ingredients, businesses and governments should evaluate the relationship between the working environment in the semiconductor/LCD companies and the incidence of various metabolic disorders as well as the CVDs which might be caused by these disorders. (page 11-12, lines 318-356)

Point 2: The authors aim seem to be evaluation of several chronic diseases. However, much of the text of the manuscript is focused on cardiovascular disease risk

Response 2: As the reviewer pointed out, we analyzed the cardiovascular disease (CVD) risks in correlation with the chronic diseases including hypertension, diabetes, and dyslipidemia which have been known as the common risk factors for the CVDs. We clarified this by revising the discussion section of the manuscript as follows:

Workers at risk of diabetes were mostly distributed in the high-risk group (67.0%, 354/528), which might mean that more attention is needed regarding the health risk of workers at risk of diabetes. The Framingham risk score was more strongly correlated with the risk levels of chronic diseases used in this study compared to the 10-year ASCVD risk score. -> Workers at risk of diabetes were mostly distributed in the high-risk group (67.0%, 354/528), which might mean that more attention is needed regarding the health risk of workers at risk of diabetes.

Since hypertension, diabetes and dyslipidemia are associated with the occurrence of CVDs significantly, we analyzed the risk of CVDs in the employees using the Framing-ham and ASCVD risk scores [31]. The Framingham risk score was more strongly correlated with the risk levels of chronic diseases used in this study compared to the 10-year ASCVD risk score. (page 13, lines 387-389)

Point 3: It is not clear what reference was used to classify risk groups. For example, on what basis (SBP/DBP) ≥ 160/100 mmHg is considered as high-risk group?

Response 3: As the reviewer pointed out, we revised the manuscript by presenting the information of  references on risk group classification as follows:

2.3. Classification of health risk groups ... We modified the company’s in-house criteria to classify risk groups in workers at risk of chronic diseases as follows. (1) The high-risk group comprised those with the risk of three of the chronic diseases (hypertension, diabetes, and dyslipidemia), those in a high-risk group in the KOSHA CVD risk assessment, those with a systolic blood pressure/diastolic blood pressure (SBP/DBP) ≥ 160/100 mmHg, or those with a body mass index (BMI) of ≥ 35 kg/m2. (2) The intermediate-risk group comprised those with the risk of two of the considered chronic diseases, those in an intermediate-risk group according to the KOSHA CVD risk assessment, those with a SBP/DBP 140−159/90−99 mmHg, or those with a BMI of 30−34 kg/m2. (3) The low-risk group was defined as those with the risk of one of the considered chronic diseases, those in a low-risk group based on the KOSHA CVD risk assessment, those with a SBP/DBP 120−139/80−89 mmHg, or those with a BMI of 25−29 kg/m2. -> We modified the company’s in-house criteria (unpublished) to classify risk groups in workers at risk of chronic diseases as follows. (1) The high-risk group comprised those with the risk of three of the chronic diseases (hypertension, diabetes, and dyslipidemia), those in a high-risk group in the KOSHA CVD risk assessment, those with a systolic blood pressure/diastolic blood pressure (SBP/DBP) ≥ 160/100 mmHg, or those with a body mass index (BMI) of ≥ 35 kg/m2. (2) The intermediate-risk group comprised those with the risk of two of the considered chronic diseases, those in an intermediate-risk group according to the KOSHA CVD risk assessment, those with a SBP/DBP 140−159/90−99 mmHg, or those with a BMI of 30−34.9 kg/m2. (3) The low-risk group was defined as those with the risk of one of the considered chronic diseases, those in a low-risk group based on the KOSHA CVD risk assessment, those with a SBP/DBP 120−139/80−89 mmHg, or those with a BMI of 25−29.9 kg/m2 [16,18,19]. (page 3, lines 143-154)

*References :

  1. Risk assessment for the prevention of cardio-cerebrovascular disease at workplace (KOSHA code H-1-2013).; Korea Occupational Safety and Health Agency: Ulsan, 2013.
  2. 2018 Hypertension Guidelines.; The Korean Society of Hypertension: Seoul, 2018.
  3. 2020 Obesity Guidelines.; Korean Society for the Study of Obesity: Seoul, 2020

Point 4: In Table 1, it hasn't been clarified what characteristics A, B, C, D, and E represent. some of the subheadings are misplaced in the middle of the table.

Response 4: According to the reviewer’s comment, we added the following description at the bottom of Table 1 and 4, and revised the overall format of the tables:

Table 1. … b Characteristics of workplaces: A and B, wafer fabrication; C, research and development; D and E, test and packaging.

Table 4. … a Characteristics of workplaces: A and B, wafer fabrication; C, research and development; D and E, test and packaging.

Round 2

Reviewer 3 Report

Song et al. did not address my first comment properly. My recommendation was to analyse their data comparing workers exposed to electronic material and waste with workers who do not have such exposures. Instead, they produced a literature review on that in the discussion section. Unless, the authors do not have such data to analyse, their response is not appropriate to my comment. The rest of the comments are addressed well and I don't have any further comments.
